# Experimental and Theoretical Study on Flexural Behavior of GFRP- and CFRP-Reinforced Concrete Beams after High-Temperature Exposure

**DOI:** 10.3390/polym14194002

**Published:** 2022-09-24

**Authors:** Jun Zhao, Haojin Pan, Zike Wang, Guanghui Li

**Affiliations:** 1School of Mechanics and Safety Engineering, Zhengzhou University, Zhengzhou 450001, China; 2Henan Transport Investment Group Co., Ltd., Zhengzhou 450016, China

**Keywords:** high-temperature exposure, GFRP bar, CFRP bar, concrete beams, bending behavior, experiment, calculation model

## Abstract

In this study, in order to study the flexural behavior of fiber-reinforced polymer (FRP) bars with reinforced concrete beams under static loads after high-temperature exposure, seven pieces of FRP-reinforced concrete beams were subjected to static bending tests and calculation model derivations. Four-point bending tests were carried out on FRP-reinforced concrete beams after high temperature treatment. The effects of high temperature and types of FRP bars on the cracking load, crack development, deflection and ultimate capacity, and failure mode of concrete beams were investigated. The test results show that the maximum crack width, deflection, and ultimate bearing capacity of GFRP- and CFRP-reinforced concrete beams decrease obviously with a rise in high temperature. After the exposure of 400 °C for 2 h, compared with the behavior of concrete beams at room temperature, the maximum crack width of GFRP and CFRP-reinforced concrete beams increased by 42.9% and 41.7%, respectively, the deflection increases by 103.6% and 22.0%, and the ultimate bearing capacity decrease by 11.9% and 3.9%. Meanwhile, through the analysis of the existing research results and test results, the calculation models for the maximum crack width, deflection, and residual ultimate capacity of FRP-reinforced concrete beams after exposure of high temperature were proposed. For FRP-reinforced concrete beams after high-temperature exposure, the errors between the measured maximum crack width, stiffness, residual bearing capacity, and their corresponding calculated values using the model were mostly less than 10%. The calculated value using the proposed model in this research is in good agreement with the measured value. The mechanical properties of FRP-reinforced high-strength concrete structures after high-temperature exposure can be preliminarily predicted, which provides a new theoretical basis for the application of FRP-reinforced concrete structures.

## 1. Introduction

The long-term performance of reinforced concrete (RC) structures has become an important design factor. In harsh environments, the corrosion problem of steel bars in RC structures cannot be ignored [1], as it can seriously affect the safety and durability of the structures [2]. In order to solve this problem, various methods have been proposed, among which fiber-reinforced polymer (FRP) composites have been more and more widely used in civil engineering as a competitive alternative to steel [3,4,5,6]. Compared to steel, FRP bars have the advantages of low weight, high tensile strength, corrosion resistance and nonmagnetic nature, high fatigue endurance, and low thermal and electrical conductivity [7,8,9]. According to the classification of fiber, the FRP used for civil engineering include carbon FRP (CFRP), glass FRP (GFRP), basalt FRP (BFRP), and aramid FRP (AFRP) bars [10]. Among them, GFRP bars are the most commonly used due to their reliable attributes and relatively low cost [11,12,13], while CFRP bars possess the most excellent mechanical properties and greatest application potential because of their high tensile strength, good fatigue performance, low density, and because they have the closest elastic modulus to concrete [14,15].

Fire is one of the most serious potential dangers when FRP-reinforced concrete structures are in service [16,17]. Therefore, it is important to investigate the behavior of FRP-reinforced concrete structures in and after fire. Clearly, the fire resistance of FRP-reinforced concrete structures mainly depends on the fire resistance of concrete [18,19], FRP bars [20], and their bonding. Due to the poor high-temperature resistance of the resin matrix in FRP bars, the degradation of FRP bars and their interface bonding with concrete after high-temperature exposure is more serious than that of concrete [21], which significantly affects the bearing capacity of the structures [20]. This paper mainly focuses on the mechanical properties of FRP-reinforced concrete beams after high-temperature exposure. So far, many scholars have studied the physical properties and tensile properties of FRP bars and bonding properties between FRP bars and concrete after high-temperature exposure [21,22,23], and the influence factors involved fiber and resin types of FRP bars [22], exposure temperature and holding time [23], protective layer thickness of bars [21], and so on. Available data indicate that at temperatures above 300 °C, the mechanical properties of GFRP and CFRP bars starts to decrease, and the tensile strength decreases faster than the elastic modulus [22]. Moreover, researchers have also found that after experiencing the same high-temperature exposure, the bond strength between the FRP bars and concrete suffered greater damage than the tensile strength of the FRP bars [22]. The bond strength between FRP bars and concrete is almost completely lost after around 300 °C exposure [24].

Using FRP bars instead of steel bars is useful for improving the durability of concrete structures [25]. However, because the elastic modulus of FRP bars and the bonding force with concrete are lower than those of steel bars, greater crack width and deflection are likely to occur in FRP-reinforced concrete beams [26]. Refai et al. [27] found that the deformability, cracking, stiffness, and bearing capacity of concrete beams reinforced with hybrid GFRP and steel bars were better than those of pure GFRP-reinforced concrete beams. El-Mogy et al. [28] and Deifalla et al. [29] considered that the addition of transverse GFRP bars reduced the deflection and increased the ultimate torsional strength of concrete beams. Mustafa [30] found that hybrid steel and CFRP-reinforced concrete beams exhibited better performance than hybrid steel and GFRP-reinforced concrete beams during crack initiation and propagation, that is, the first crack of hybrid GFRP-reinforced concrete beams appeared at both the midspan and middle support simultaneously, while the cracks of hybrid CFRP-reinforced concrete beams were only found at mid-span. Pecce [31] proposed a modified ACI model and MGL model for calculating deflection and crack width. The modified ACI model fits well with the deflection of FRP-reinforced concrete beams. Moreover, the MGL model required a proper evaluation of the bond coefficient *k*_f_. Mojtaba [32] proposed a formulation to calculate the effective flexural stiffness (EFS) of FRP-reinforced concrete beams, and the effects of FRP longitudinal bar ratio, effective depth-to-height ratio, elastic modulus of FRP, and concrete compressive strength on the EFS ratio were studied.

Because of the poor heat resistance of FRP bars, the flexural performance of FRP-reinforced concrete beams at high temperatures is different from that at room temperatures. Lin and Zhang [33] studied the flexural performance of FRP-reinforced concrete beams exposed to ISO R834 standard fire using the finite element method, and found that since the strength and elastic modulus of CFRP bars decreased less than that of GFRP or AFRP bars, the deflection of CFRP-reinforced concrete beams during the heating period is smaller than that of GFRP or AFRP-reinforced concrete beams. Further, Yu and Kodur [34] predicted the fire response of concrete beams reinforced with FRP bars by numerical simulation. The study results showed that GFRP-reinforced concrete beams have lower fire resistance than CFRP-reinforced concrete beams. Saafi [35] proposed an estimation method to easily calculate the residual flexural and shear capacities of FRP-reinforced concrete beams at high-temperature exposure according to the properties of FRP bars and concrete at high temperature. The author also found that the flexural and shear capacities-degradation of GFRP-reinforced concrete beams with the increase of temperature is more obvious than those of CFRP beams. Rafi and Nadjai [36] experimentally analyzed the fire-resistance performance of CFRP and/or steel bars-reinforced concrete beams. The results showed that with the prolongation of heating time, the predicted deflection of the beam using the orthogonal cracking model was in better agreement with the experimental data. Rafi et al. [37,38,39] also experimentally studied the fire resistance of CFRP-reinforced concrete beams. The cracking behavior of CFRP-reinforced concrete beams was similar to that of steel-reinforced beams at an elevated temperature. In addition, steel-reinforced concrete beams are destroyed due to steel yielding, while CFRP beams failed by concrete crushing.

However, the existing research on the flexural performance of FRP-reinforced concrete beams after high-temperature exposure is still relatively limited; further experimental and theoretical research on this aspect is needed. Therefore, this paper developed an experimental program and a calculation method to study the flexural performance of FRP-reinforced concrete beams after high-temperature exposure. The impact parameters adopted in this research were high temperature levels (200, 400, 600 °C) and FRP types (GFRP, CFRP). The cracking load, crack development, deflection, residual flexural bending, and failure mode of FRP-reinforced concrete beams after different high-temperature exposure were measured. Finally, the calculation method of maximum crack width, deflection, and ultimate bending capacity of FRP-reinforced concrete beams after high-temperature exposure is proposed, providing a theoretical basis for whether FRP-reinforced concrete structures can continue to serve after fire, so as to promote the development of FRP-reinforced concrete structures.

## 2. Experimental Program

In this research, seven FRP-reinforced concrete beams were tested by a four-point loading setup. In these beam specimens, two beams were statically loaded at room temperature (25 °C), while the other beams were loaded after different high-temperature exposure treatments (200, 400, and 600 °C, respectively, for 2 h). For simplicity, the letters G and C denote the GFRP and CFRP beam specimens, respectively, and the figures 25, 200, 400, and 600 denote the different exposure temperatures. Accordingly, “G-25” refers to an unheated GFRP beam specimen loaded at 25 °C. Similarly, “C-400” refers to a CFRP beam specimen after high-temperature exposure at 400 °C.

### 2.1. Test Specimens

The design of the beams was based on ACI 440.1R-15 [40] and GB 50608-2010 [41]. Four beams were GFRP bars-reinforced concrete beams and three were CFRP beams. The grouping situation of the test plan is shown in Table 1. The specimen size and reinforcement details are shown in Figure 1a. All seven beams were 1800 mm long rectangular beams with a cross-sectional dimension of 150 mm × 200 mm. There were two 12 mm GFRP longitudinal bars or 10 mm CFRP bars in the bottom tension zone of the test beam, and two 10 mm discontinuous HRB 335 bars in the top compression zone. Each beam specimen was transversely strengthened with 8 mm diameter HPB 235 steel stirrups, and the stirrup center spacing was 100 mm.

### 2.2. Materials

#### 2.2.1. Concrete

The beams were manufactured using identical concrete mixes. The coarse and fine aggregates were composed of continuous grading gravel with grain size of 5 to 20 mm, respectively, and river sand with a fineness module of 3.0. Cementitious material was ordinary Portland cement (PO 42.5), and a polycarboxylic acid water-reducing agent was used to enhance the workability of concrete. Concrete mix proportion design is consistent with the author’s previous research [42], and the performance of concrete after high temperature is detailed in the author’s previous research [43]. Based on GB/T50081-2019 [44], the slump of concrete ranged from 140 to 160 mm. The average 28-day cube and prism compressive strengths of concrete were 66.9 MPa and 59.7 MPa, respectively, and the modulus of elasticity of the concrete was 34.7 GPa.

#### 2.2.2. FRP Bars

The GFRP bars and CFRP bars used were the same as in the author’s previous research [42], and the mechanical properties of GFRP bars after elevated temperatures can be found in the author’s previous research [45]. According to ASTM D7205/D7205M-06 (2016) [46], the tensile strength, elastic modulus, and other physical properties of GFRP and CFRP bars were determined and are given in Table 2.

Moreover, the thermal properties of FRP bars were also evaluated by dynamic mechanical thermal analysis (DMTA) and thermogravimetric analysis (TGA). The instruments and test modes used can be found in the author’s previous research [42]. The DMTA results of FRP bars are shown in Figure 2. At 25 °C, the storage modulus of GFRP bars is significantly lower than that of CFRP bars, but with the increase in temperature, the storage modulus of the two became gradually close. Inversely, the glass transition temperature (*T*_g_) of CFRP (139 °C) was lower than that of GFRP (155 °C).

The TGA results can be found in the author’s previous research [42]. In both air and argon, the mass loss of both GFRP and CFRP bars essentially starts to decrease at 300 °C. The resin of the GFRP bar is basically completely carbonized when temperature reaches 400 °C, and the quality is no longer reduced; within a high temperature of 800 °C, the high temperature has basically no effect on the quality loss of the glass fiber. While in air, due to the reaction of carbon fiber and oxygen, when the temperature is higher than 600 °C, the quality of CFRP bars is significantly reduced, and the quality of CFRP bars is almost completely lost at 800 °C.

### 2.3. Furnace Equipment and Instrumentation

The five test beams were heated using a gas-electric mixed furnace purchased from Changsha Kehui Furnace Industry Science and Technology Co., Ltd. (Changsha, China); as shown in Figure 3a, the furnace’s internal heating chamber measures 2200 mm × 700 mm × 600 mm. As shown in Figure 3b, beam specimens were heated according to the experimental fire curve controlled by a computer program. According to the heating treatment schedule (200 °C for 2 h, 400 °C for 2 h, and 600 °C for 2 h), the furnace was heated to the target temperature and held for 2 h. The temperature measured at the location of FRP bars after heating is shown in Figure 3c. After heating, the specimens were cooled in the air for testing.

Four-point bending tests were conducted on seven beam specimens using 100 kN electro-hydraulic servo static and dynamic actuators purchased from Jinan Sans Dynamic Testing Technology Co., Ltd. (Jinan, China) The schematic diagram of test setup and photograph are shown in Figure 1b and c, respectively. Concrete strain development at different beam depths was measured using five strain gauges (S2120-100AA, Hebei Xingtai Jinli Sensor factory, Xingtai, China) with lengths of 100 mm. Five linear variable displacement transducers (LVDTs) (YHD-100, Liyang City Instrument and Meter Plant, Liyang, China) were placed below the pure bending zone and above the support of the beam, and a high-speed static data logger (uT7130, Wuhan uTekL Electronic Technology Co., Ltd., Wuhan, China) was used to record strain and deflection data. The crack width development was measured by a crack observation (ZBL-F130, Beijing Zhongbo Technology Co., Ltd., Beijing, China) with a precision of 0.02 mm.

### 2.4. Test Procedure

All beams were simply supported with a net span of 1500 mm and tested under four-point bending, and the loading procedure of the test was in accordance with GB/T 50152-2012 [47]. The loading method is graded loading. Each load stage was 2 kN before the appearance of the first crack and 4 kN after the first crack. The monitoring of cracks, concrete strain, and mid-span deflections continued over the entire loading process. When the bearing capacity of the test beam decreased rapidly due to the crushing of concrete in the compression zone or the fracture or pull-out of FRP bars, the load ends. Schematic diagrams and photographs of FRP-reinforced concrete beams under static load are shown in Figure 1b and c, respectively.

## 3. Results and Discussion

### 3.1. Cracking Behavior and Failure Modes

#### 3.1.1. Crack Development Analysis

The distribution of cracks when the test beams G-25, G-200, G-400, G-600, C-25, C-400, and C-600 failed are shown in Figure 4. The figures on the beams in Figure 4 represent the sequence of cracks. Except for the test beam C-600, the crack height of the other test beams slowly developed with the increase of load. Bending cracks in the range of loading positions propagated vertically upward toward the compressed concrete zone. The cracks outside the loading position developed vertically upward at the initial stage of cracking, and gradually developed to the loading point position with the increase in load. This is similar to the crack development in the research of Rabee et al. [48]. The mean crack spacings of test beams G-25, G-200, G-400 and G-600 were 180 mm, 143 mm, 128 mm, and 120 mm, respectively. Compared with the corresponding GFRP-reinforced concrete beams at the same temperature, the mean crack spacings of C-25 and C-400 decreased, and the crack spacings were 130 mm and 110 mm, respectively. During the whole loading process, only one crack with a width greater than 2 mm was produced in the specimen C-600. After cracking, even under the same load value, the crack width of C-600 had been increasing; eventually the CFRP bars were pulled out of the concrete. The other test beams were damaged because of the fracture of FRP bars at the main crack.

#### 3.1.2. Failure Modes Analysis

The failure modes of FRP-reinforced concrete beams under load are basically the same as those at room temperature. However, due to the low reinforcement ratio of FRP bars, the concrete in the compression zone of test beams did not reach the ultimate compressive strength during the loading process, so no concrete compression failure occurred. This is consistent with the Kostiantyn’s research [49], and due to the adverse effect of high temperature on the properties of FRP bars, all the beam specimens after high-temperature exposure failed in rupturing of FRP bars. The failure modes of FRP beams include bending failure [27] (FRP bars fracture failure) and debonding failure between FRP bars and concrete. For bending failure, the test beam first presented vertical upward cracks in the mid-span of the loading position interval. With the increase of load, new cracks appeared continuously, and the width of existing cracks gradually increased. When the load increased to (0.5~0.6) ultimate load (Fu), the cracks on the test beams appeared almost completely. When the load reached (0.8~0.9) Fu, the cracks accelerated upward and the crack height approached the loading point. Finally, the FRP bars were broken (specimens G-25, G-200, G-400, G-600, C-25 and C-400). The typical fracture failure mode of FRP bars is shown in Figure 5a.

The failure mode of the specimen C-600 was the debonding failure between FRP bars and concrete. When the load reached 2 kN, the specimen cracked, and then the crack width expanded. The CFRP bars were quickly pulled out from the concrete at the crack. Typical debonding failure between FRP bars and concrete is shown in Figure 5b. The test result of internal temperature distribution of concrete beams (Figure 3c) shows that the actual temperature of FRP bars in CFRP-reinforced concrete beams was 525 °C when exposed to 600 °C for 2 h. Previous research [29,45] has shown that internal cohesive stress between concrete is almost completely lost at this temperature. Meanwhile, the TGA results show that the quality retention rate of CFRP bars after burning in argon at 525 °C is 80.4%, and the quality loss is obvious. At this time, the bonding effect of resin in CFRP fails, and the longitudinal carbon fiber is in a loose state, which leads to the complete loss of bonding between CFRP bars and concrete [50].

### 3.2. Cracking Load

The cracking load (*P*_cr_) and moment (*M*_cr_) of FRP-reinforced concrete beams after different high temperatures are shown in Table 3. The comparison of cracking moments of FRP-reinforced concrete beams after different high-temperature exposure is shown in Figure 6. As shown in Table 3 and Figure 6, high temperature has a significant impact on the *M*_cr_ of FRP beams, and *M*_cr_ decreases with the increase of temperature. According to the relevant research results [38], the *M*_cr_ of concrete beams is mainly determined by the tensile strength of concrete and the beam section size. After high-temperature exposure, the loss of tensile strength of concrete is much larger than that of compressive strength, and after 400 °C, the tensile strength decreases sharply [51]. In this experiment, the influence of high temperature on the deterioration of tensile strength of concrete was significant, and lead to a significant reduction in the *M*_cr_ of FRP-reinforced concrete beams after high-temperature exposure [35]. In addition, as shown in Figure 6, compared to the specimens at 25 °C, after high-temperature exposure of 400 °C and 600 °C, *M*_cr_ of CFRP beams decreased more than that of GFRP beams; this is related to the material behavior of CFRP bars exposed to high temperatures (above 400 °C). Previous research [52] has shown that the tensile failure of CFRP bars exposed to temperatures of 400 °C is due to slippage from the anchorage rather than the damage of the tendon itself. The loss of epoxy resin means that the load cannot be transferred effectively in FRP bars. After 500 °C exposure, all the epoxy resin of the FRP bar evaporated, and only a bunch of loose fibers remained.

### 3.3. Concrete Strain

The change of concrete strain (*ε*_c,mid_) in the mid-span section of test beams under different loads after high-temperature exposure is shown in Figure 7. From Figure 7, it can be seen that for the test beams at 25 °C and after high-temperature exposure, when the load is small (before concrete cracking), the strain of concrete in the mid-span section is well in line with the plane section assumption, and after concrete cracking, it is also approximately in line with the plane section assumption. Since the strain gauge of the concrete in the tensile zone failed after cracks appeared in the mid-span of the test beams, the strain of the concrete in the tensile zone under load after failure was not collected. By comparing the concrete strain of the compression zone in Figure 7, it can be found that at the same load, with the increase of exposure temperature, the compressive strain of the concrete in the compression zone of test beams increased gradually and the strain after high-temperature exposure increased significantly compared to that at room temperature.

### 3.4. Maximum Crack Width Analysis

The variation of maximum crack width (*ω*_max_) in GFRP and CFRP beams after different high-temperature exposure is shown in Figure 8a and b, respectively. Meanwhile, the effect of FRP types on variation of *ω*_max_ with the load of concrete beams is shown in Figure 8c.

From Figure 8a and b, it can be seen that the crack width of GFRP beam cracking at 25 °C is large, and approaches the limit of 0.5 mm specified in GB50608-2010 [41]. With the increase of exposure temperature, the cracking load of GFRP beams decreases. The *ω*_max_ increases linearly with the rise of load and the increase rate of it rises gradually with the augment of high temperature. The *ω*_max_ of specimen C-400 is obviously larger than that of specimen C-25. When the load is constant, due to the impact of high temperature, the strength and elastic modulus of FRP bars and concrete decrease [22,53], the bonding force between FRP bars and concrete is destroyed, and the bond slip between them increases [29], which in turn leads to the increase of crack width of test beams after high-temperature exposure.

In addition, the type of FRP bars has a significant impact on the *ω*_max_ of concrete beams, as shown in Figure 8c. When the load and temperature condition are the same, the *ω*_max_ of CFRP beams is significantly smaller than that of GFRP beams. At 25 °C and 400 °C, the maximum crack widths of GFRP beams are 55.7% and 51.0% larger than those of CFRP beams, respectively.

### 3.5. Load-Deflection Behavior

The load mid-span deflection curves of GFRP and CFRP beams after different high temperatures exposure are shown in Figure 9a and b, respectively.

At 25 °C, the mid-span deflection curve of FRP beam is basically approximate to the double broken line. Before cracking, FRP bars and concrete bear the tensile force together. The stiffness of the test beam is large, and the slope of the mid-span deflection curve is large. After cracking, the stiffness of FRP beam decreases significantly, and the slope of the mid-span deflection curve decreases and the curve has an obvious inflection point [27]. Since the tensile stress-strain curve of FRP bars before fracture is linear, the relation curve between load and mid-span deflection of the test beams at the stage from cracking to failure is also basically linear. In the later stage of loading, due to the fact that the nonlinearity of concrete is gradually obvious, the load-deflection curve of the mid-span of the beam also shows a certain degree of nonlinearity. For GFRP beams at 400 °C and 600 °C, due to the small cracking load, there is no obvious inflection point in the relation curve between load and mid-span deflection, which is completely similar to the straight line in shape. Moreover, the slope of the curve decreases with the increase of temperature, which indicates that the bending stiffness of GFRP beams decreases significantly after high-temperature exposure.

It can be seen from Figure 9b and c that before concrete cracking, the deflections of CFRP and GFRP beams are basically the same, and the type of FRP bars has little effect on the stiffness of concrete beams. However, after concrete cracking, under the same load, the mid-span deflection of GFRP beams is significantly higher than that of CFRP beams, for example, the mid-span deflection is 9.7% and 44.6% higher at 25 and 400 °C, respectively. The type of FRP bars has a significant impact on the deflection of FRP beams, this is mainly due to the fact that the elastic modulus of GFRP bars is significantly lower than that of CFRP bars (see in Table 2), so the deformation of GFRP bars and deflection of GFRP beams after loading is larger than those of CFRP bars and beams. Kostiantyn’s research [49] showed similar results; the deflection of HFRP-RC (HFRP: hybrid composite of carbon fibers and basalt fibers bars) beams after high-temperature exposure is about half of that of BFRP-RC beams due to the difference in elastic modulus between carbon and basalt fibers.

### 3.6. Bending Capacity Analysis

The bending capacity of FRP-reinforced concrete beams is shown in Table 4. With an increase of temperature, the bending capacity of GFRP beams decreases significantly. At 200, 400, and 600 °C, the ultimate load (*F*_u_) of GFRP beams is 0.97, 0.89 and 0.24 times of that of GFRP beams at 25 °C, respectively. It can be concluded that high temperature has a significant impact on the bending capacity of GFRP beams. The ultimate load of CFRP beams decreased slightly at 400 °C, which was 0.96 times of that at 25 °C. This indicates that when the temperature is not higher than 400 °C, the deterioration of bending capacity of CFRP beams is relatively weak. Compared with the test beams at 25 °C, the reduction rate of the ultimate moment (*M*_u_) of GFRP beams at 400 °C was higher than that of CFRP beams, which were 11.0% and 4%, respectively. The reason is that the degradation of mechanical properties of GFRP bars exposed to temperature less than 400 °C is greater than that of CFRP bars [35]. However, when the exposure temperature reaches 600 °C, the bending capacity of CFRP beams is almost completely lost due to serious debonding of CFRP bars to concrete, which is much lower than that of GFRP beams.

## 4. Theoretical Analysis

### 4.1. Calculation Method of Maximum Crack Width ω_max_

#### 4.1.1. Calculation Method of *ω*_max_ of FRP-Reinforced Concrete Beam at 25 °C

In this section, two calculation formulas are used to calculate the *ω*_max_ of GFRP- and CFRP-reinforced concrete beams, and the accuracy of the two models is evaluated by comparing the calculation results with the measured results. The formulas given in GB 50608-2010 [41] are as follows:(1)$$\omega_{\max}=2.1\psi \frac{\sigma_{f}{}_{k}}{E_{f}}(1.9c+0.08\frac{d_{\mathrm{eq}}}{\rho_{te}})$$
(2)$$\psi =1.1-0.65\frac{f_{tk}}{\rho_{te}\sigma_{\mathrm{fk}}}$$
(3)$$d_{\mathrm{eq}}=\frac{\sum n_{i}d_{i}^{2}}{\sum n_{i}\nu_{i}d_{i}}$$
(4)$$\rho_{\mathrm{te}}=\frac{A_{f}}{A_{\mathrm{te}}}$$
(5)$$\sigma_{\mathrm{fk}}=\frac{M_{k}}{0.90A_{f}h_{0f}}$$

The modified formula for calculating the maximum crack width of FRP-reinforced concrete beams is as follows:(6)$$\omega_{\max}=2.1\psi \frac{\sigma_{fk}}{E_{f}}(1.9c+0.08\frac{d_{\mathrm{eq}}}{\rho_{te}})$$
(7)$$\psi =1.3-0.74\frac{f_{\mathrm{tk}}}{\rho_{\mathrm{te}}\sigma_{\mathrm{fk}}}$$
(8)$$d_{\mathrm{eq}}=\frac{\sum n_{i}d_{i}^{2}}{\sum n_{i}\nu_{i}d_{i}}$$
(9)$$\rho_{\mathrm{te}}=\frac{A_{f}}{A_{\mathrm{te}}}$$
(10)$$\sigma_{\mathrm{fk}}=\frac{M_{k}}{0.90A_{f}h_{0f}}$$

The meaning of each symbol in the two sets of formulas is consistent with Formula (6.2.2) in Chapter 6 of GB 50608-2010 [41].

The comparison between the measured maximum crack width of FRP-reinforced concrete beams at 25 °C and the calculated values using above two formulas is summarized in Table 5. It can be seen that the GB 50608-2010 calculated value *ω*_max1_ is greatly different from the measured value. The modified calculated value *ω*_max2_ is in good agreement with the measured value.

GB 50010-2010 [54] is calculated according to the calculation method of reinforced concrete beams, and the calculation formula of the uneven tension coefficient *ψ* of FRP longitudinal bars between cracks is given as follows:(11)$$\psi =S_{1}\times (1-0.8\frac{M_{\mathrm{cr}}}{M})$$
where *M*_cr_ is the cracking moment of the component section, *M* is the bending moment acting on the component section during crack calculation, and *S*_1_ is the coefficient and is taken as 1.1. When the bending moment *M* is equal to the cracking moment *M*_cr_, the component is in a critical state, and at this time the calculated value of *ψ* is 0.22. When the calculated value of *ψ* is less than 0.2, *ψ* is taken as 0.2.

However, some scholars have conducted research and discussed whether taking *S*_1_ as 1.1 is suitable for FRP-reinforced concrete beams. Zhu [55] obtained the calculation formula of *ψ* by analyzing the relevant test data and suggesting that the coefficient *S*_1_ should be 1.3, as shown in Equations (12) and (13). When the component is in a critical state, the calculated value of *ψ* is 0.26. In addition, research [56,57] has found that the initial crack width of FRP-reinforced concrete beams is larger, and the number of cracks is less than, those of steel-reinforced concrete beams due to the relatively lower bond strength between FRP bars and concrete. The tensile force borne by the concrete in the tension zone of FRP beam is smaller than that of the reinforced concrete beam, and the strain unevenness of the FRP bars is smaller. Therefore, when the calculated value of *ψ* is less than 0.3, *ψ* is suggested to be taken as 0.3.
(12)$$\psi =1.3\times (1-0.8\frac{M_{\mathrm{cr}}}{M})$$
(13)$$0.8\frac{M_{\mathrm{cr}}}{M}=0.57\frac{f_{tk}}{\rho_{te}\sigma_{\mathrm{fk}}}$$

#### 4.1.2. Calculation Method of *ω*_max_ of FRP-Reinforced Concrete Beam after High-Temperature Exposure

The test results show that the cracking load of FRP beams decreases significantly with the increase of exposure temperature. It can be seen from Equation (7) that with the decrease of the cracking load *f*_tk_ of FRP beams, the uneven tensile coefficient *ψ* gradually increases. At the same time, due to the influence of high temperature, the elastic modulus of FRP bars and the bonding strength between FRP bars and concrete decreases [24], and the maximum crack width of FRP beams increases. In view of this, the influence coefficient of high-temperature exposure on the maximum crack width *K*_w_ related to high temperature is introduced. In order to keep consistent with the calculation method of the current code, based on the bond-slip theory and combined with GB 50608-2010 [41], it is recommended that the calculation model of the maximum crack width of FRP-reinforced concrete beams after high-temperature exposure is taken as Equation (14). At 600 °C, the bending capacity of GFRP beam is obviously deteriorated. The ultimate load of specimen G-600 is only 24.0% of that of specimen G-25. However, specimen C-600 completely loses its bending capacity and its maximum crack width increases exponentially. Considering the maximum crack width of specimen G-600 is no longer of practical significance, 400 °C is taken as the high temperature limit in this study. Based on the test data and the safety design of structure, the calculation formula of *K*_w_ is taken as Equation (15).
(14)$$\omega_{\max,T}=K_{w}\omega_{\max}$$
(15)$$K_{w}=1.668(\frac{T}{1000})+0.958\;25\leq T\leq 400{\;}^{o}C$$
where *ω*_max,*T*_ is the maximum crack width of FRP beam after high-temperature exposure of *T* °C, *ω*_max_ is the maximum crack width calculated according to Equation (6), *K*_w_ is the influence coefficient of high-temperature exposure on the maximum crack width of FRP beam, determined by experiment, and *T* is the temperature of high-temperature exposure.

The calculated and measured maximum crack widths of GFRP and CFRP beams at 200 °C and 400 °C are shown in Table 6. It can be seen that the ratio of the measured value to the calculated value of *ω*_max,*T*_ of the FRP beam after high-temperature exposure is close to 1, and the error is basically less than 10%. The calculated values are in good agreement with the measured values.

### 4.2. Calculation Method of Deflection

#### 4.2.1. Short-Term Stiffness *B_s_* of FRP-Reinforced Concrete Beams at Room Temperature

In this section, two formulas are used to calculate the short-term stiffness and mid-span deflection of GFRP-and CFRP-reinforced concrete beams, and the accuracy of the two models is evaluated through comparing the calculation results with the measured results. The short-term stiffness formula of FRP-reinforced concrete beams under standard combined load given in GB 50608-2010 [41] is as follows:(16)$$B_{s}=\frac{E_{f}A_{f}h_{0f}^{2}}{1.15\psi +0.2+\frac{6\alpha_{\mathrm{fE}}\rho_{f}}{1+3.5\gamma_{f}^{\prime }}}$$
where *ψ* is calculated according to Equation (2).

The modified calculation formula of short-term stiffness is as follows:(17)$$B_{s}=\frac{E_{f}A_{f}h_{0f}^{2}}{1.11\psi +0.2+\frac{6\alpha_{\mathrm{fE}}\rho_{f}}{1+3.5\gamma_{f}^{\prime }}}$$
where *ψ* is calculated according to Equation (7), *γ*_f_^′^ is the ratio of the area of the compression flange to the effective area of the web, $\gamma_{f}^{\prime }=\frac{(b_{f}^{\prime }-b)h_{f}^{\prime }}{bh_{0f}}$, *b*_f_^’^ and *h*_f_^’^ are the width and height of the compression flange respectively. *ρ*_f_ is the reinforcement ratio of the FRP bars in the longitudinal tension, and for the flexural members of the FRP bars, $\rho_{f}=A_{f}/bh_{0f}$, *α*_fE_ is the ratio of FRP bar elastic modulus to concrete elastic modulus, $\alpha_{\mathrm{fE}}=E_{f}/E_{c}$, *σ*_fk_ is the stress of FRP bars under load effect standard combination, $\sigma_{\mathrm{fk}}=\frac{M_{k}}{0.9A_{f}h_{0f}}$.

In order to simplify the calculation, the “minimum stiffness principle” is used to calculate the deflection of FRP-reinforced concrete flexural beams. The calculated span is *l*, and the mid-span deflection of a simply supported beam, which bears two concentrated loads of *P/2* at 1/3 its span, can be obtained by Equation (18):(18)$$f=\frac{6.81Pl^{3}}{384E_{c}I_{e}}$$
where *f* is the mid-span deflection, *P* is the load, *l* is the span of the beam, *E*_c_ is the elastic modulus of the concrete, *I*_e_ is the effective moment of inertia of the cross section.

Equations (16) and (17) are used to calculate *B*_s_, and it is substituted into Equation (18) to get the calculated value of the mid-span deflection of the beam. The calculation results of FRP-reinforced beams at 25 °C are shown in Figure 10. It can be seen from Figure 10 that the measured mid-span deflection of GFRP and CFRP beams is close to the calculated value before concrete cracking. After concrete cracking, the measured and calculated values of mid-span deflections of GFRP and CFRP beams are quite different. The calculated deflection using Equation (16) is slightly low and insecure. Research [55] has modified the short-term stiffness calculation formula of FRP beams according to experimental research and mechanism analysis. The short-term stiffness test results of FRP-reinforced beams at 25 °C are shown in Table 7, and the deflection results of them are shown in Figure 10.

It can be seen that the mean ratio of the measured short-term stiffness to the modified calculated value is significantly higher than GB 50608-2010 calculated value. This indicates that the modified short-term stiffness is closer to the test results. It can be seen from Figure 10 that the calculated mid-span deflection using Equation (17) is also more consistent with the measured value than that using Equation (16).

#### 4.2.2. Short-Term Stiffness of FRP-Reinforced Concrete Beam after High-Temperature Exposure

The test results show that under the same load, the deflection of FRP beam increases gradually with the augment of exposure temperature, indicating that its stiffness decreases. The effect of high temperature reduces the tensile strength of concrete, the elastic modulus of FRP bars, and the bond strength between FRP bars and concrete. Meanwhile, the uneven tensile coefficient of FRP longitudinal bars between cracks increases, resulting in the decrease of the stiffness of test beams. Therefore, when calculating the deflection of an FRP beam after high-temperature exposure, the short-term stiffness reduction factor *K*_b_ of FRP beam is introduced. Relevant research [58] shows that the degradation degree of different types of FRP bars is different at high temperature. Considering the influence of FRP bar type, the influence coefficient *λ*_f_ of FRP bar type is introduced. The calculation model of short-term bending stiffness of FRP-reinforced concrete beams after high-temperature exposure is shown in Equation (19). At the same time, through the regression analysis of the test results, the calculation formula of stiffness reduction coefficient *K*_b_ of FRP-reinforced concrete beams after high-temperature exposure is obtained, as shown in Equation (20). The influence coefficient *λ*_f_ of FRP bar type is determined. For GFRP bars, $\lambda_{f}=1.0$, and for CFRP bars $\lambda_{f}=1.10$.
(19)$$B_{s,T}=\lambda_{f}K_{b}B_{s}$$
(20)$$K_{b}=\frac{1}{3.13(\frac{T}{1000})+0.922}25\leq T\leq 400{\;}^{o}C$$
where *B*_s_ is the short-term stiffness of FRP-reinforced concrete beams at 25 °C calculated by Equation (17).

Table 8 shows the comparison between the actual measured value and the calculated value of the short-term stiffness *B*_s,_*_T_* of the FRP beam after high-temperature exposure. From the data in the Table 8, the mean ratio of the measured short-term stiffness to the calculated value of Equation (19) is close to 1, which proves that the calculated value is in good agreement with the actual test value.

### 4.3. Calculation Method of Bending Capacity of Cross Section of FRP-Reinforced Concrete Beams after High-Temperature Exposure

The GB 50608-2010 [41] gives the calculation method of bending capacity of normal section of FRP-reinforced concrete beams at room temperature. Based on this, this section calculates the residual bending capacity of FRP-reinforced concrete beams after high-temperature exposure, and compares it with the measured value of the test.

#### 4.3.1. Balanced Relative Depth of Compressive Area *ξ*_fb,*T*_ after High-Temperature Exposure

After high-temperature exposure, the tensile strength and elastic modulus of FRP bars deteriorate, which leads to the change of *ξ*_fb,*T*_ of the FRP-reinforced concrete beams. The ultimate tensile strength of FRP longitudinal tension bars and the crushing of concrete in compression zone occur simultaneously when the critical failure occurs in FRP- reinforced concrete beams after high-temperature exposure. Referring to the calculation formula of *ξ*_fb,*T*_ at room temperature given in GB 50608-2010 [41], *ξ*_fb,*T*_ of FRP beams after high-temperature exposure should be calculated according to Equation (21):(21)$$\xi_{\mathrm{fb},T}=\frac{\beta_{1}\varepsilon_{\mathrm{cu}}}{\varepsilon_{\mathrm{cu}}+f_{\mathrm{fu},T}/E_{f,T}}=\frac{\beta_{1}}{1+\frac{f_{\mathrm{fu},T}}{\varepsilon_{\mathrm{cu}}E_{f,T}}}$$
where *ξ*_fb,*T*_ is the relative boundary compressive region’s height of FRP beams after high-temperature exposure of *T* °C, *β*_1_ is the equivalent rectangular stress diagram coefficients, which are implemented in accordance with the provisions of Section 6.2.6 of GB 50010-2010 [54]. *Ε*_cu_ is the ultimate compressive strain of normal cross-section concrete at room temperature, which is taken as 0.0033, *f*_fu,*T*_ is the ultimate tensile strength of FRP bars after high-temperature exposure of *T* °C, calculated from Formula (4.3) in Chapter 4 of the author’s doctoral thesis [59]. *E*_f,*T*_ is the tensile elastic modulus of FRP bars after high-temperature exposure of *T* °C, calculated from Formula (4.4) in Chapter 4 of the author’s doctoral thesis [59].

#### 4.3.2. Calculation Formula of Residual Bending Capacity of Cross Section after High-Temperature Exposure

Figure 11 shows the equivalent stress diagram for the calculation of the normal section bending capacity of FRP-reinforced concrete beams of rectangular cross-section. Due to the linear elastic properties of FRP bars, GB 50608-2010 [41] and related research results [60,61] suggest that the design of FRP-reinforced concrete beams should refer to over-reinforced reinforced concrete beams. Due to the different high-temperature conditions, there may be two failure modes of FRP-reinforced concrete beams after high-temperature exposure. Therefore, two cases are considered respectively in the calculation of bending capacity of FRP-reinforced concrete beams after high-temperature exposure as follows:

(1) When $x\leq \xi_{\mathrm{fb},T}h_{0f}$, FRP bar fracture failure occurs in the FRP beam in the tension zone. At this time, the concrete in the compression zone does not reach the ultimate compressive strain or compression failure occurs. The residual bending capacity is mainly determined by the design value of the tensile strength of the FRP bar after high-temperature exposure, and the residual bending capacity of the normal section should be calculated according to Equation (22).
(22)$$M\leq f_{\mathrm{fu},T}A_{f}(h_{0f}-\frac{x}{2})$$
(23)$$x=\frac{f_{\mathrm{fu},T}A_{f}}{f_{c,T}b}$$

(2) When $x>\xi_{\mathrm{fb},T}h_{0f}$, the FRP beam has a crushing failure mode of concrete in the compression zone. At this time, the FRP bars in the tension zone have not reached the ultimate tensile strength, and the residual bending capacity of the normal section should be calculated according to Equation (24).
(24)$$M\leq \alpha_{1}f_{c,T}bx(h_{0f}-x/2)$$
(25)$$x=\xi_{\mathrm{fb},T}h_{0f}$$
where *f*_c,_*_T_* is the value of the axial compressive strength of the concrete after the high temperature of *T* °C, calculated from Formula (3.2) in Chapter 3 of the author’s doctoral thesis [59], *f*_cu,_*_T_* is the ultimate tensile strength of FRP bars after high-temperature exposure of *T* °C, calculated from Formula (4.3) in Chapter 4 of the author’s doctoral thesis [59], *A*_f_ is the cross-sectional area of FRP rebar, *ξ*_fb,_*_T_* is the relative boundary compressive region’s height of FRP beams after high-temperature exposure of *T* °C, calculated according to Equation (21), *b* is the cross-section width of the beam, *h*_0f_ is the distance between joint point of FRP bar and the top surface of the concrete beam, *x* is the concrete compression zone height after high-temperature exposure.

#### 4.3.3. Analysis of Calculation Results

The bounding relative height of compression zone and the calculated failure mode of FRP-reinforced concrete beams after high-temperature exposure are shown in Table 9. It can be seen from Table 9 that the failure mode of FRP-reinforced concrete beams after high-temperature exposure calculated by Equation (21) is consistent with the actual failure mode, and the failure mode of FRP- reinforced concrete beams after high-temperature exposure are all FRP ruptures.

The Equation (22) is used to calculate the residual bending capacity of FRP-reinforced concrete beams after high-temperature exposure, and the calculation results are shown in Table 10. It is noted that when at 600 °C, the reduction amplitude of ultimate moment of normal section of GFRP-reinforced concrete beams is 76%, and the bending capacity of CFRP beams has been completely lost. Therefore, from the perspective of safety, it can be considered that the bending capacity of cross section of FRP-reinforced concrete beams is completely lost at 600 °C, and its bending capacity is no longer considered in the calculation. At 200 °C and 400 °C, the ratio of measured and calculated values of GFRP beams is 0.850 and 1.135, respectively, and that of CFRP beams is 1.162. The error between *M*_exp_ and *M*_c_ of FRP-reinforced concrete beams after high-temperature exposure is about 15%. On the whole, the calculated value is in good agreement with the measured value.

## 5. Conclusions

This study describes the four-point bending test phenomena of FRP-reinforced concrete beams after different high-temperature exposures, and analyzes the influence of high temperature on the cracking moment, crack development, failure mode, maximum crack width, deflection, and residual bending capacity of FRP-reinforced concrete beams. The main conclusions are as follows:(1)The maximum crack width of FRP-reinforced concrete beams deteriorates obviously after high-temperature exposure. With an increase of exposure temperature from 25 °C to 600 °C, the maximum crack width of FRP beams increases gradually. The type of FRP bars has a significant influence on the maximum crack width of FRP-reinforced concrete beams. After exposure of 25 °C and 400 °C, the maximum crack width of GFRP-reinforced concrete beams is 55.7% and 51.0% higher than that of CFRP beams, respectively.(2)High temperature deteriorates the deformation resistance of GFRP-reinforced concrete beams obviously. With an increase in high temperature, the stiffness of FRP-reinforced concrete beams decreases obviously and the deflection increases. The type of FRP bars has a significant effect on the deflection of FRP-reinforced concrete beams. At 25 °C and 400 °C, the deflection of GFRP-reinforced concrete beams is 26.9% and 44.5% higher than that of CFRP beams, respectively.(3)The effect of high temperature on the bending capacity of the normal section of FRP-reinforced concrete beams is obvious. With the increase of high temperature, the bending capacity of the normal section of FRP beams decreases significantly. Especially at 600 °C, the bending capacity of the normal section of GFRP-reinforced concrete beams decreases to 76%, which was 4.5 kN·m, and the bending capacity of the normal section of CFRP beams is completely lost, only 0.5 kN·m.(4)Based on the analysis of the existing research results and test results, the calculation models and methods of the maximum crack width, deflection, short-term stiffness, and residual bending capacity of FRP-reinforced concrete beams, considering the influence of high temperature are proposed, respectively. Compared with the calculation results of the existing model, it is found that the modified model is in better agreement with the measured value.

This study provided research on the flexural behavior of FRP-reinforced concrete beams after high-temperature exposure. In the future, research will be extended to investigate the mechanical properties of FRP-reinforced concrete beams at elevated temperatures.

## Figures and Tables

**Figure 1 polymers-14-04002-f001:**
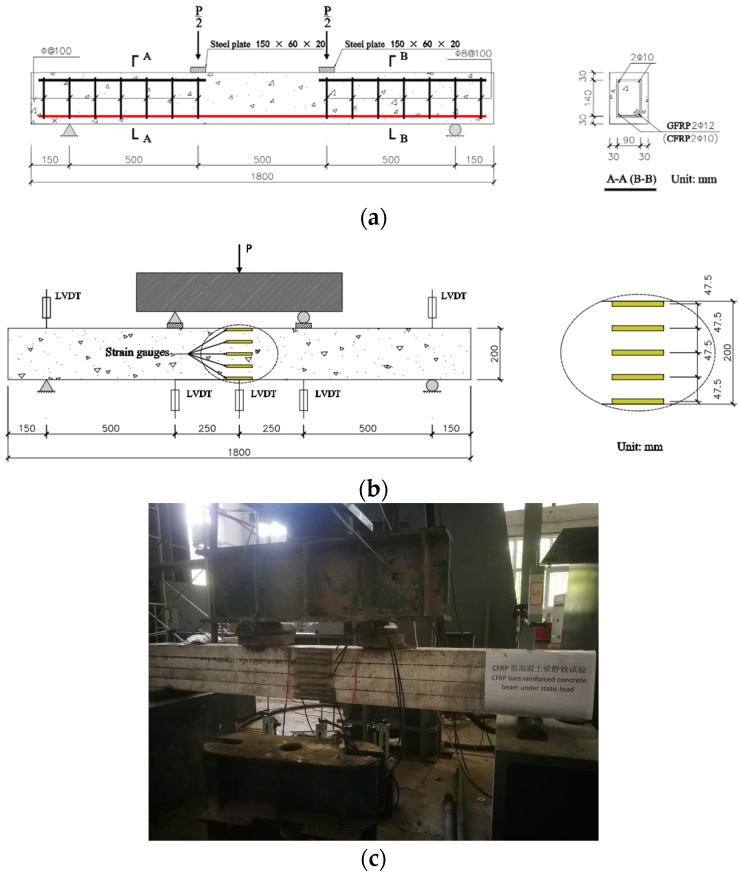
Specimen and test setup: (**a**) specimen size and reinforcement (note: 30 mm concrete cover for top, bottom, and side); (**b**) schematic diagram; (**c**) photograph.

**Figure 2 polymers-14-04002-f002:**
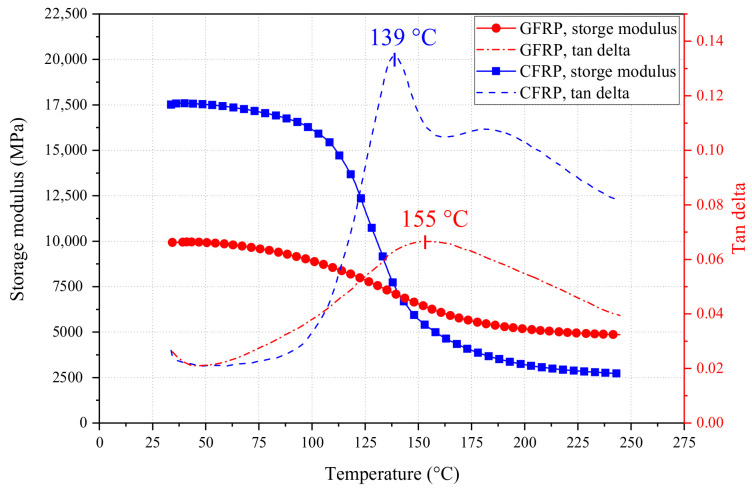
DMTA curves of GFRP and CFRP bars.

**Figure 3 polymers-14-04002-f003:**
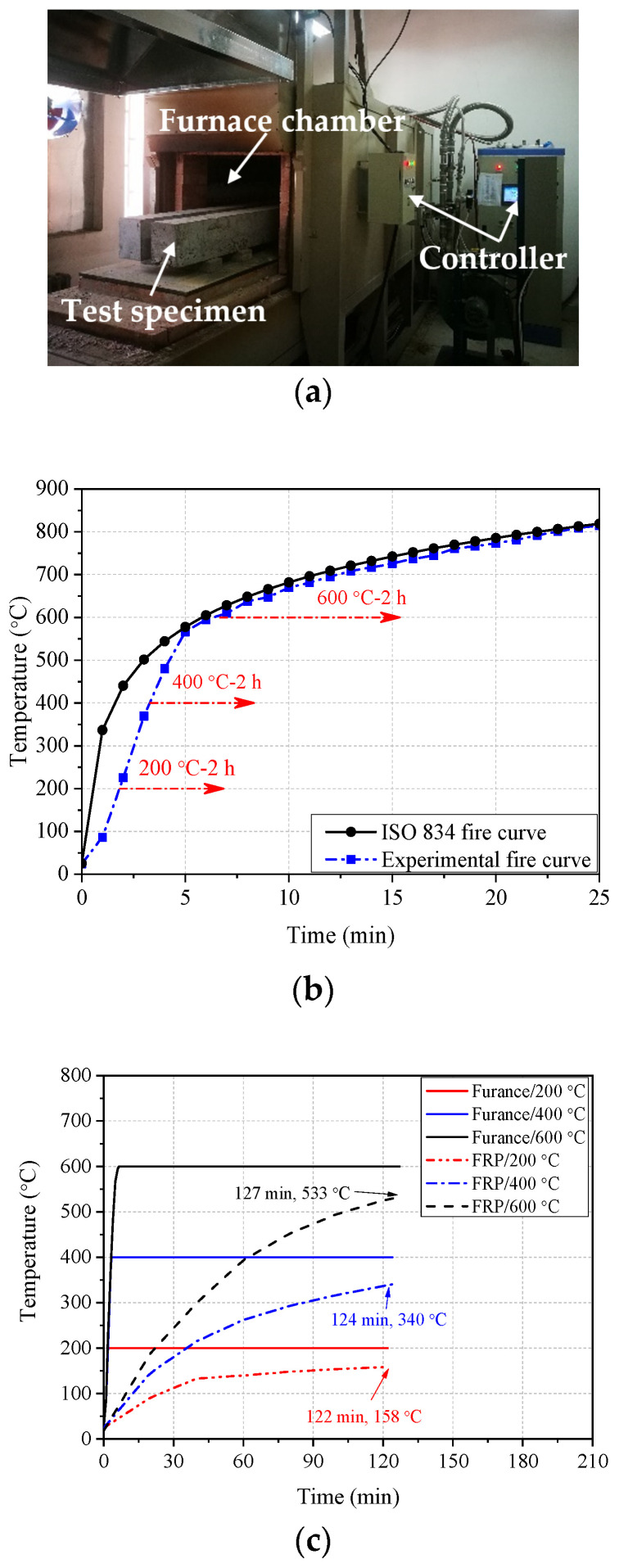
Heating system for test beams: (**a**) furnace; (**b**) heating scheme for test specimens; (**c**) measured temperature at the location of FRP bars.

**Figure 4 polymers-14-04002-f004:**
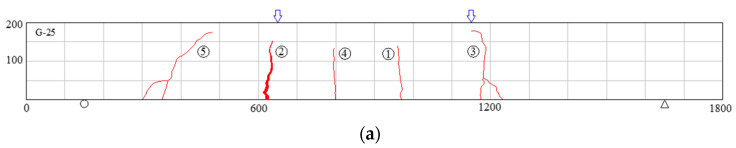
Crack distributions in all tested beams when fail (note: the figure in the circle represents the occurrence sequence of cracks, the arrows above the beam represent the loading points): (**a**) G-25; (**b**) G-200; (**c**) G-400; (**d**) G-600; (**e**) C-25; (**f**) C-400; (**g**) C-600.

**Figure 5 polymers-14-04002-f005:**
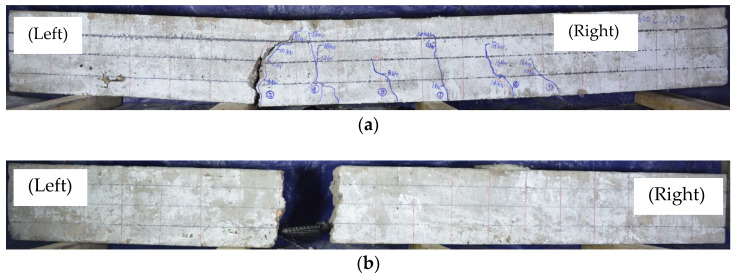
Two kinds of typical failure modes of tested concrete beams: (**a**) GFRP rupture (e.g., G-600); (**b**) CFRP debonding (e.g., C-600).

**Figure 6 polymers-14-04002-f006:**
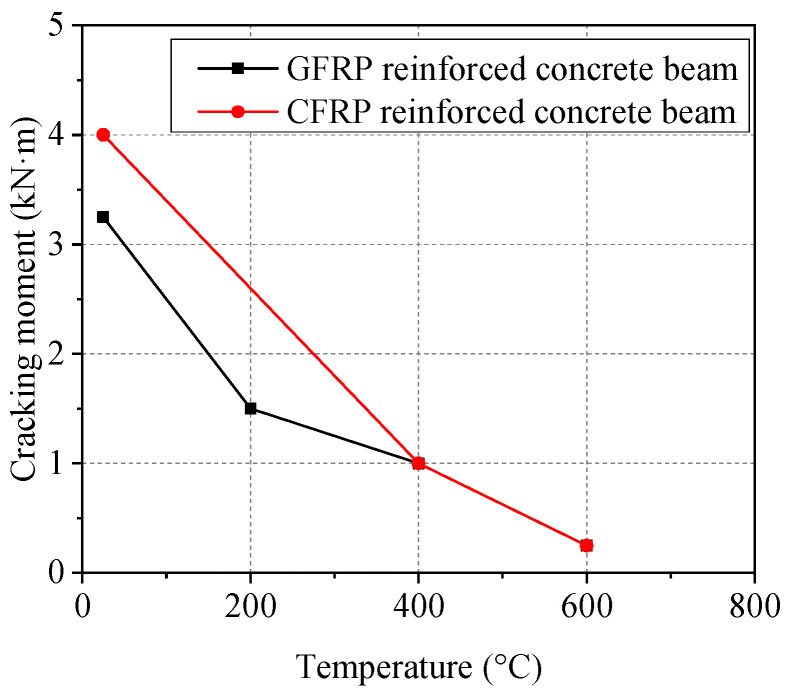
Cracking moment of FRP-reinforced concrete beams after high-temperature exposure.

**Figure 7 polymers-14-04002-f007:**
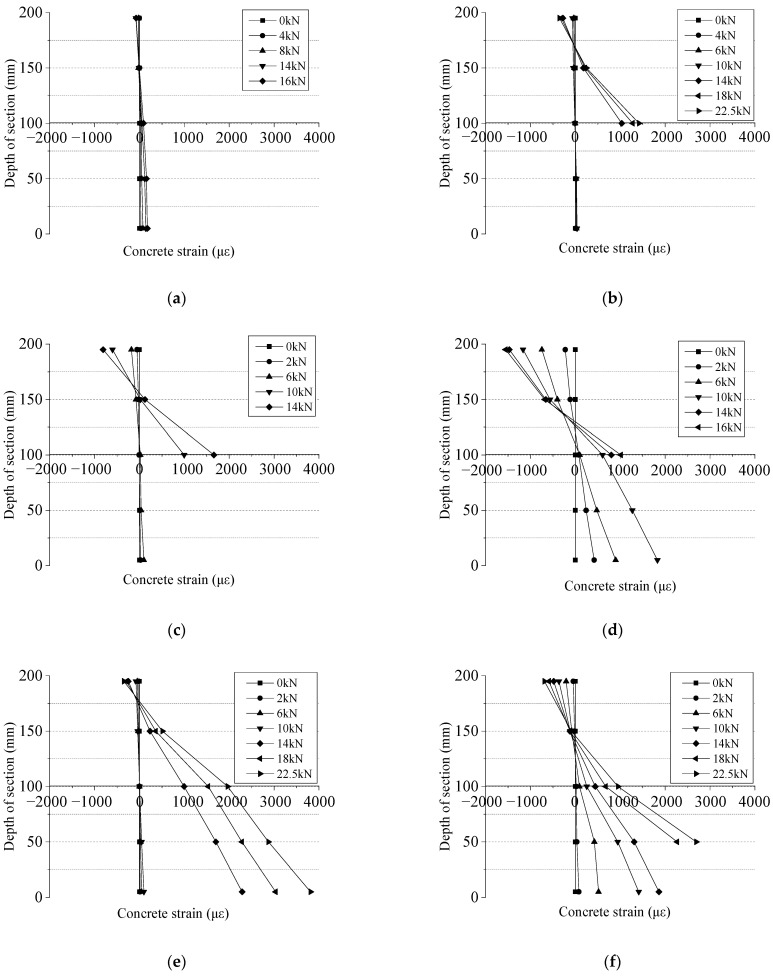
The concrete strain distribution along the section depth at the mid-span section of the tested beams: (**a**) G-25; (**b**) G-200; (**c**) G-400; (**d**) G-600; (**e**) C-25; (**f**) C-400.

**Figure 8 polymers-14-04002-f008:**
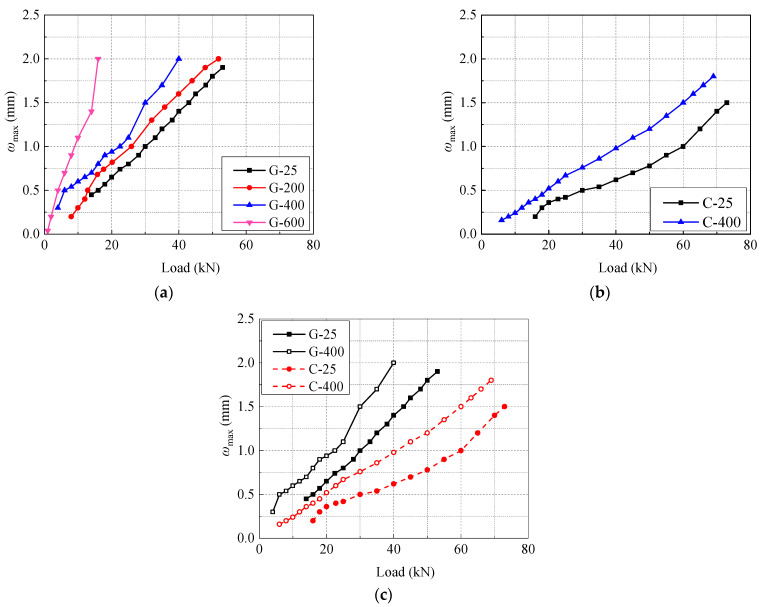
Developments of maximum crack width (*ω*_max_) of beams with different FRP bar types after elevated temperature exposure: (**a**) effect of temperature of GFRP bar; (**b**) effect of temperature of CFRP bar; (**c**) effect of FRP bar type.

**Figure 9 polymers-14-04002-f009:**
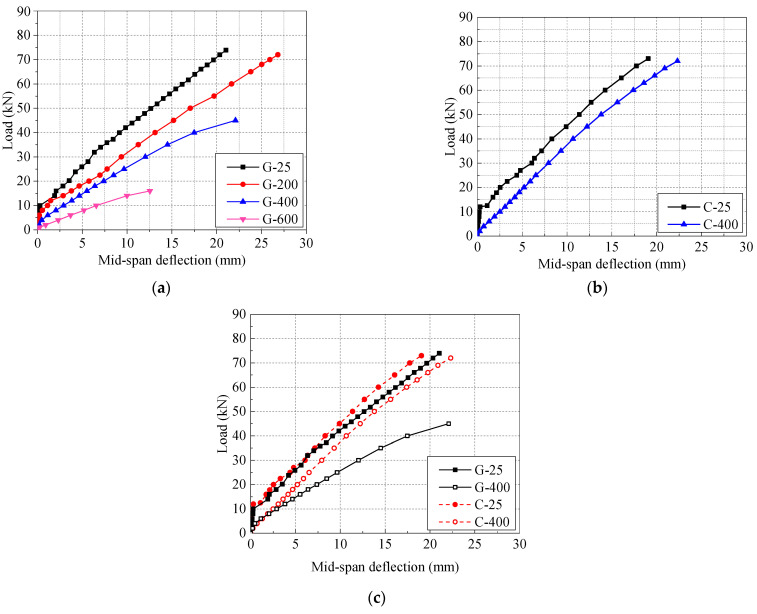
Load-deflection relationship for concrete beams with different FRP bar types after elevated temperature exposure: (**a**) effect of temperature of GFRP bar; (**b**) effect of temperature of CFRP bar; (**c**) effect of FRP bar type.

**Figure 10 polymers-14-04002-f010:**
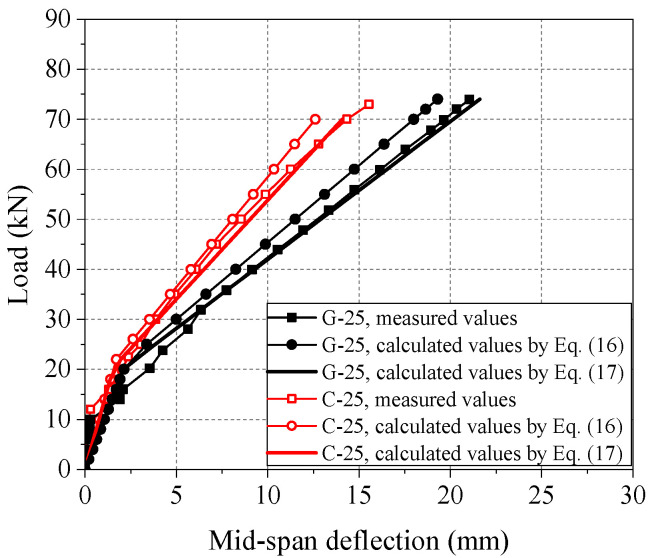
Comparison between measured and calculated values of mid-span deflection of FRP-reinforced concrete beams at room temperature.

**Figure 11 polymers-14-04002-f011:**
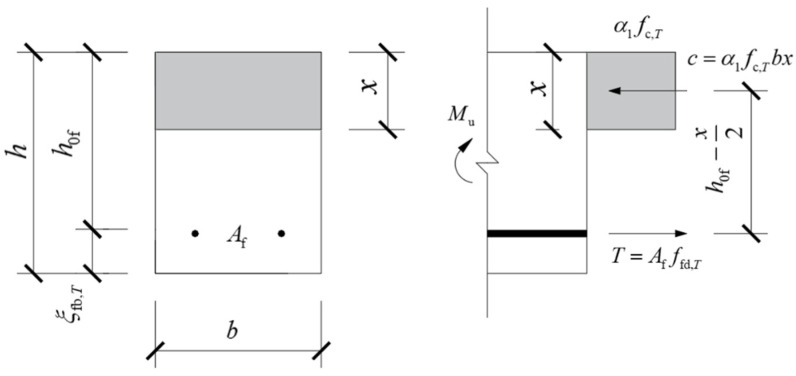
Equivalent stress diagram for cross-section bearing capacity of FRP-reinforced concrete beams after high-temperature exposure.

**Table 1 polymers-14-04002-t001:** Summary of all FRP-reinforced concrete beam specimens used in this research.

Specimen ID	FRP Bar Type	Diameter (mm)	Exposure Temperature (°C)	Holding Time (h)
G-25	GFRP	12	25	-
G-200	GFRP	12	200	2
G-400	GFRP	12	400	2
G-600	GFRP	12	600	2
C-25	CFRP	10	25	-
C-400	CFRP	10	400	2
C-600	CFRP	10	600	2

Note: “-” represents “not available”.

**Table 2 polymers-14-04002-t002:** Tensile and physical properties of G/CFRP bars used in this research.

	Properties	GFRP Bar	CFRP Bar
Tensile property	Ultimate strength (MPa)	810 ± 33	1238 ± 39
	Modulus of elasticity (GPa)	47 ± 2	117 ± 4
	Ultimate strain (%)	1.72 ± 0.05	1.06 ± 0.01
Physical property	Nominal diameter (mm)	12	10
	Nominal area (mm^2^)	113	79
	Density (g/cm^3^)	2.07	1.51
	Fiber content by volume (%)	64.9	55.8
	Fiber content by weight (%)	79.8	65.1
	Glass transition temperature ^a^ *T*_g_ (°C)	155	139
	Longitudinal coefficient of thermal expansion (×10^−6^/°C) [40]	6.0~10.0	−9.0~0.0
	Transverse coefficient of thermal expansion (×10^−6^/°C) [40]	21.0~23.0	74.0~104.0

Note: ^a^: The peak value of tan delta curve was used as the glass transition temperature.

**Table 3 polymers-14-04002-t003:** Static test results of tested concrete beams.

Specimen ID	Cracking Load *P*_cr_ (kN)	Cracking Moment *M*_cr_ (kN·m)	Maximum Crack Width *ω*_max_ (mm)	Failure Mode
G-25	13.0	3.25	1.9	GFRP rupture
G-200	6.0	1.5	2	GFRP rupture
G-400	4.0	1.0	2	GFRP rupture
G-600	1.0	0.25	2	GFRP rupture
C-25	16.0	4.0	1.5	CFRP rupture
C-400	4.0	1.0	1.8	CFRP rupture
C-600	1.0	0.25	-	Debonding

Note: “-” represents “not available”.

**Table 4 polymers-14-04002-t004:** Bending capacity of FRP-reinforced concrete beams.

Specimen ID	Ultimate Load *F*_u_ (kN)	Ultimate Moment *M*_u_ (kN·m)	$\frac{F_{\mathrm{high}\;\mathrm{temperature}}}{F_{\mathrm{room}\;\mathrm{temperature}}}$	Ultimate Moment Decreases Amplitude (%)
G-25	75	18.75	1	-
G-200	73	18.2	0.97	3
G-400	67	16.8	0.89	11
G-600	18	4.5	0.24	76
C-25	76	19.0	1	-
C-400	73	18.25	0.96	4
C-600	2	0.5	0.03	97

Note: “-” represents “not available”.

**Table 5 polymers-14-04002-t005:** Comparison between measured and calculated values of maximum crack width of tested beam specimens at room temperature.

Specimen ID	Load (kN)	Measured Value *ω*_exp_ (mm)	GB 50608-2010 Calculated Value *ω*_max1_ (mm)	Modified Calculated Value *ω*_max2_ (mm)	$\frac{\omega_{\exp}}{\omega_{\max1}}$	$\frac{\omega_{\exp}}{\omega_{\max2}}$
G-25	25	0.8	0.40	0.51	2.0	1.569
G-25	30	1.0	0.67	0.84	1.493	1.190
C-25	25	0.42	0.24	0.30	1.75	1.40
C-25	30	0.50	0.40	0.50	1.25	1.0

**Table 6 polymers-14-04002-t006:** Comparison between measured and calculated values of maximum crack width of tested beam specimens after high-temperature exposure.

Specimen ID	Load (kN)	Measured Value *ω*_exp_ (mm)	Calculated Value *ω*_max,*T*_ (mm)	$\frac{\omega_{\exp}}{\omega_{\max,T}}$
G-200	32	1.30	1.25	1.04
G-200	36	1.45	1.58	0.918
G-400	30	1.50	1.36	1.103
G-400	35	1.70	1.88	0.904
C-400	30	0.76	0.81	0.938

**Table 7 polymers-14-04002-t007:** Comparison between measured and calculated values of short-term stiffness of tested beam specimens at room temperature.

Specimen ID	Load (kN)	Measured Deflection (mm)	Measured Value *B_s,_*_exp_ (×10^11^·N·mm^2^)	GB 50608-2010 Calculated Value *B*_s1_ (×10^11^·N·mm^2^)	$\frac{B_{s,}{}_{\exp}}{B_{s}{}_{1}}\;$	Modified Calculated Value *B*_s2_ (×10^11^·N·mm^2^)	$\frac{B_{s,}{}_{\exp}}{B_{s}{}_{2}}$
G-25	32	6.345	3.01	3.39	0.888	3.00	1.003
G-25	40	9.148	2.62	2.90	0.901	2.58	1.014
G-25	50	12.646	2.36	2.60	0.908	2.32	1.019
G-25	60	16.147	2.22	2.43	0.912	2.17	1.022
C-25	30	3.870	4.64	5.96	0.779	5.31	0.874
C-25	40	6.080	3.94	4.87	0.810	4.35	0.906
C-25	50	8.550	3.50	4.39	0.797	3.93	0.891
C-25	60	11.270	3.19	4.11	0.776	3.69	0.864

**Table 8 polymers-14-04002-t008:** Comparison between measured and calculated values of short-term stiffness *B*_s,*T*_ of tested beam specimens after high-temperature exposure.

Specimen ID	Load (kN)	Measured Deflection (mm)	Measured Stiffness *B*_exp_ (×10^11^·N·mm^2^)	Calculated Value *B*_s,*T*_ (×10^11^·N·mm^2^)	$\frac{B_{\exp}}{B_{s,T}}$
G-200	30	9.364	1.92	1.99	0.966
G-200	40	13.124	1.83	1.61	1.133
G-200	50	17.054	1.76	1.45	1.213
G-400	30	12.026	1.49	1.43	1.043
G-400	35	14.512	1.44	1.26	1.141
G-400	40	17.478	1.37	1.16	1.181
G-400	45	22.090	1.22	1.09	1.117
C-400	30	7.929	2.27	2.68	0.847
C-400	40	10.670	2.25	2.20	1.023
C-400	50	13.804	2.17	1.99	1.090

**Table 9 polymers-14-04002-t009:** Calculated $\xi_{\mathrm{fb},T}h_{0f}$, *x* values and failure modes of tested beam specimens after high-temperature exposure.

Specimen ID	$\xi_{\mathrm{fb},T}h_{0f}$	x	Calculated Failure Mode	Actual Failure Mode
G-200	27.1	20.0	FRP rupture	FRP rupture
G-400	33.0	17.6	FRP rupture	FRP rupture
C-400	48.2	18.7	FRP rupture	FRP rupture

**Table 10 polymers-14-04002-t010:** Comparison between measured and calculated values of bending capacity of tested beam specimens after high-temperature exposure.

Specimen ID	Measured Ultimate Moment *M*_exp_ (kN·m)	Calculated Ultimate Moment *M*_c_ (kN·m)	$\frac{M_{\exp}}{M_{c}}$
G-200	18.2	21.4	0.850
G-400	16.8	14.8	1.135
C-400	18.25	15.7	1.162

## Data Availability

The data presented in this study are available on request from the corresponding author.

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
