# Peer review of "Experimental and Theoretical Study on Flexural Behavior of GFRP- and CFRP-Reinforced Concrete Beams after High-Temperature Exposure"

_polymers, 2022, doi:10.3390/polym14194002_

Round 1

Reviewer 1 Report

This manuscript studied the flexural behavior of FRP bars reinforced concrete beams under static loads after high-temperature exposure. The results showed that after the exposure of 400oC for 2h, compared with the behavior of concrete beam at room temperature,  the deflection increased by 103.6% and 22.0%, and the ultimate bearing capacity decreased by 11.9%, and 3.9% of GFRP and CFRP reinforced concrete beams, respectively. The main drawbacks of this research are:

1-      Mechanical properties of individual materials at elevated temperatures, such as concrete which contains gravel which is more susceptible to temperature, and FRP bars which have different volume fractions of resin, did not study. See, https://doi.org/10.3390/polym14030472

2-      The beams were subjected to high-temperature exposure under zero-Load, which is not illogical and does not happen in reality.

These drawbacks must be discussed and highlighted, and it will be better if the authors overcome one or both of these drawbacks. This manuscript needs essential modifications before it can be accepted for publication as follows:

·         The English language and structure of the manuscript must be improved.

·         The results presented in Figs. 2 and 3 in the present manuscript were previously published by the authors (See Ref. [40], Figs 2 & 4). This discrepancy must be removed.

·         Figure 4 must be removed or corrected. The crack patterns usually crack length with the applied load, NOT crack number at the ultimate load. See Fig. 7 in https://doi.org/10.3151/jact.2.419

·         The last three raws must be deleted in Tables 6 to 8 and 10. Calculating the mean, S.D, and C.V% for such ratios for different beams is meaningless.

·         The analysis should be improved and compared with others.

Reviewer 2 Report

The paper "Experimental and theoretical study on flexural behavior of GFRP- and CFRP-reinforced concrete beams after high temperature exposure" is generally well presented and written, but it presents problems that must be evaluated by the authors:

(a) The topic is widely discussed in the literature and has approaches already known by the international scientific community. Authors should answer clearly, and add in the text, what is their real innovation?

(b) The abstract needs to have a clearer conclusion, showing the scientific contribution of this study!

(c) The introduction could be improved, some topics related to reinforcement of cementitious materials, their transition zones and alternative materials should be inserted, consider the following papers: 10.1007/s12649-021-01374-5; 10.1016/j.cscm.2022.e01273; 10.1016/j.cscm.2022.e00896.

(d) Exposure temperatures and specimen dimensions must be justified.

(e) Some images need to be improved for the internal text;

(f) Discussions need to be further discussed based on comparisons with other studies in the international literature.

Reviewer 3 Report

Manuscript ID: polymers-1915498

Title: Experimental and theoretical study on flexural behavior of GFRP- and CFRP-reinforced concrete beams after high temperature exposure

Journal: Polymers

Comments to authors:

The present study aims to investigate the flexural behavior of fiber reinforced polymer (FRP) bars reinforced concrete beam under static loads after high temperature exposure. The static flexural test of 7 FRP bar reinforced concrete beams after high temperature exposure are conducted. In the test, the effects of high temperature and types of FRP bars on the cracking load, crack development, deflection and ultimate capacity and failure mode of concrete beams are investigated. This is well-written manuscript with substantial novelty. However, to improve the quality of the manuscript, please address the following comments:

1)      Please highlight the novelty in the Abstract.

2)      English proofreading is required for some grammatical mistakes and typos for instance, ‘2200mm × 700mm × 600mm’ should be as ‘2200 mm × 700 mm × 600 mm’ i.e., space between numerical values and units.

3)      Some information about the experimental methodology and problem statement should be highlighted before the results section in the Abstract.

4)      No need to explain all the results in Abstract.

5)      The novelty and significance of the present work should be highlighted in the last paragraph of the Introduction section.

6)      The authors are recommended to add latest relevant literature review on such works.

7)      What is the need for this work? For which practical applications, the present work is helpful?

8)      The literature review should be improved by adding latest references and discussion.

9)      Thermal properties of both types of FRP bars should be added to Table 2.

10)  ‘Results and Discussions’ should be written as ‘Results and Discussion’.

11)  Work methodologies need more discussion.

12)  Finite element modeling is also recommended along with the theoretical analysis.

13)  More discussion on the post-peak behavior of fabricated beams is required (although authors used load-control technique for tests).

14)  Conclusions should be refined and elaborated. More numerical results should be added.

15)  Future recommendations can be added.

Round 2

Reviewer 1 Report

The authors have successfully addressed all my comments.  Therefore, I recommend the publication of this manuscript.

Reviewer 2 Report

Ok